# Learning Patient Rotation Using Synthetic X-ray Images from 3D CT Volumes

**Wai Yan Ryana Fok**[1,2]                    RYANA.FOK@SIEMENS-HEALTHINEERS.COM
**Andreas Fieselmann**[1]                    ANDREAS.FIESELMANN@SIEMENS-HEALTHINEERS.COM
**Magdalena Herbst**[1]                    MAGDALENA.HERBST@SIEMENS-HEALTHINEERS.COM
**Dominik Eckert**[1,2]                    DOMINIK.ECKERT@SIEMENS-HEALTHINEERS.COM
**Marcel Beister**[1]                    MARCEL.BEISTER@SIEMENS-HEALTHINEERS.COM
**Steffen Kappler**[1]                    STEFFEN.KAPPLER@SIEMENS-HEALTHINEERS.COM
[1] *X-ray Products, Siemens Healthcare GmbH, Forchheim, Germany*
**Sylvia Saalfeld**[2]                    SYLVIA.SAALFELD@OVGU.DE
[2] *Faculty of Computer Science, Otto-von-Guericke-University Magdeburg, Germany*

## Abstract

Deep learning has become a standard method for pattern recognition in medical images, but curation of large-scale annotated clinical data is challenging due to scarcity or ethical issues. Alternatively, synthetically generated data could supplementary be used to train neural networks. In this work, we propose the novel training scheme that uses synthetic chest X-rays generated from 3D photon-counting CT volumes for quantifying the internal patient rotation $\alpha$. This can automatically inform the technician if and how re-exposure is needed without the need of extensive image analysis. X-ray images were forward projected with a step size of $2°$ rotation along patient axis. 1167 images and labels were trained on a modified DenseNet-121 to detect $\alpha$. Results on 252 test images showed good correlation between true and predicted $\alpha$, with $R^2 = 0.992$, with 95% confidence level of $\approx \pm 2°$. [1]

**Keywords:** Synthetic Data, Patient Rotation Detection, Photon-counting CT, Chest X-ray

## 1. Introduction

Chest X-ray (CXR) is one of the most frequently acquired medical images. The preferred setup is posterior-anterior (PA) CXR, where the patient is standing in front of the detector. However, for immobile patients, only anteroposterior (AP) CXR can be performed, where the detector is positioned behind the patient on the bed. It is not uncommon that the patient is rotated due to sickness or medical instruments. This rotation could lead to changes in lung density and trachea position, thus reducing diagnostic confidence. Currently, cardiothoracic ratio and the clavicle-spine distance are used to determine if a CXR is rotated. However, such evaluation might require clinical expertise and hinder clinical workflow. Hence, an algorithm to quantify internal patient rotation is desired, which can automatically inform the technician if and how the re-exposure is needed.

There is an emerging usage of realistic synthetic data for machine learning in medicine (Chen et al., 2021). Synthetic medical data generated by forward simulated models (Fok

---

1. The work presented in this paper is not commercially available.

et al., 2022), physical simulations (Moturu and Chang, 2018) or AI-driven generative models, helped improving learning performance. A CNN trained with synthetic X-ray using CT-derived airspace quantification achieved expert radiologist level of accuracy on real CXR (Barbosa Jr et al., 2021). Synthetic X-rays from generative networks (GAN) were used for lesion segmentation, landmark detection and surgical tool detection learning (Gao et al., 2023), which outperformed real-data-trained models due to the effectiveness of training on a larger dataset. However, GANs could be vulnerable to generalization and may fail to reproduce anatomically accurate images (Yi et al., 2019).

In this study, we trained a network to estimate patient rotation using synthetic CXR generated by forward projecting from photon-counting CT volumes. Our proposed approach enables us to generate projections from different angles of the same CT volume, thus allowing for the automatic generation of a large amount of training CXR and ground truth labels at the same time. Moreover, these projections closely resemble real CXR as they are generated from patient CT volumes. We hypothesize that the trained model would implicitly learn features in chest rotation without the need for annotations such as cardiothoracic ratio or clavicle-spine distance.

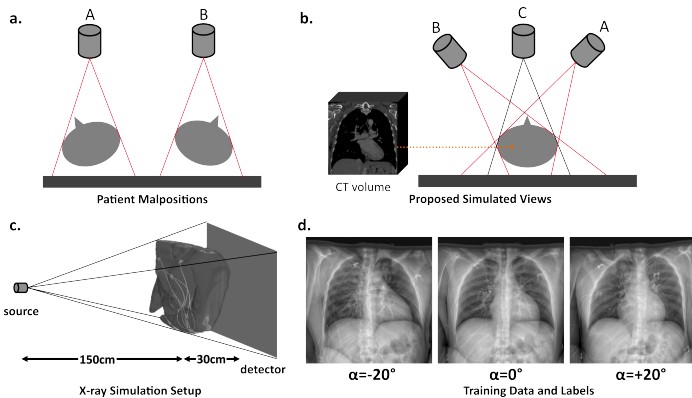

Figure 1: (a) Patient rotation along z-axis; (b) Proposed simulation of rotated (A,B) and non-rotated (C) image; (c) Forward projection setup; (d) Examples of synthetic chest X-rays with no rotation 0°, and maximum rotation at -20° and 20°.

## 2. Methods and Materials

**Synthetic X-ray Generation**  A total of 80 photon-counting CT datasets were used, each with voxel size $0.5{\times}0.5{\times}0.7\text{mm}^3$, and $\approx$1000 slices. Each CT volume underwent forward projection by ray tracing, which takes into account the cone-beam geometry of the system. X-rays are projected with angle $\alpha$ in range of [-20°, 20°], with a step size of 2° and the central projection at 0°. The X-ray source to patient distance is 150 cm, patient to detector distance is 30 cm, and the simulated detector is 1800×1800 pixels. Furthermore, standard radiographic image post-processing and cropping to the lung region were applied. X-ray simulation illustration and examples of generated images are shown in Figure 1.

**Network and Experiment**   A total of 1680 synthetic X-ray images were generated from 80 patients, each with 21 projections. Training, validation and testing consist of 1176, 252, and 252 images, respectively. All images were resized to 256×256 pixels, and intensities normalized to [0, 1]. We used DenseNet-121 (Huang et al., 2017). We used hyperbolic tangent function (Tanh) as the activation function in the final output layer, so to preserve the sign as our target labels consist of negative and positive values. We also map the output values to the range of [-20, 20]. The model was trained on Nvidia RTX A40 GPU with batch size of 16. Mean-squared error loss and Adam optimizer with learning rate of 0.01 were used and early stopping at epoch 203.

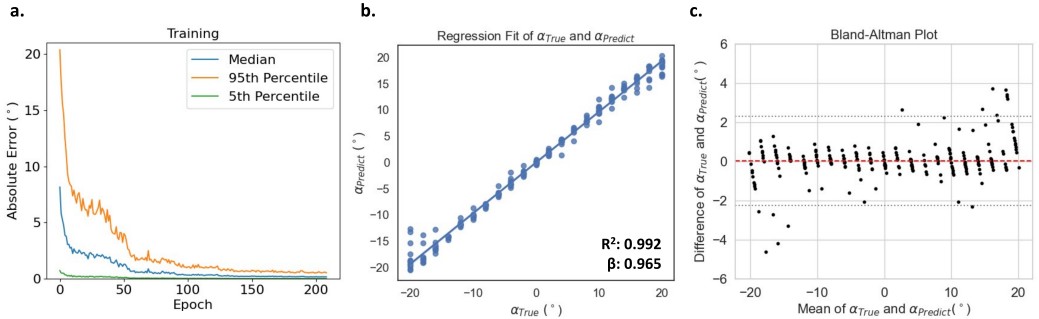

Figure 2: (a) The absolute error between $\alpha_{predict}$ and $\alpha_{true}$ along training epochs; (b) Regression fit for $\alpha_{predict}$ and $\alpha_{true}$ in test data with coefficient of determination $R^2$ and linear slope coefficients $\beta$; (c) Bland-Altman plot for the differences of $\alpha_{predict}$ and $\alpha_{true}$ in test data. Red dashed line indicates mean difference, gray dotted lines indicate 95% confidence interval.

## 3. Results and Discussion

From Figure 2a, the median, $5^{th}$ and $95^{th}$ percentile of absolute error between $\alpha_{predict}$ and $\alpha_{true}$ level off around zero after $\approx$ 150 epochs in training. On the test data (n = 252), the regression fit (Figure 2b) shows the range of prediction. Diagonal line and $R^2 = 0.992$ indicate good correlation between $\alpha_{predict}$ and $\alpha_{true}$. In Figure 2c, the differences between $\alpha_{predict}$ and $\alpha_{true}$ scattered evenly across the mean difference = 0.0385°, and close to the zero line, which shows no bias. Most data points lies within the 95% confidence interval limits (mean $\pm$ 1.96 $\times$ standard deviation of the differences) at $-2.25°$ and $2.33°$, which agrees well as our synthetic X-ray images were simulated with a 2° step size. This also indicates no systematical error in synthetic X-ray generation and the modeling of this learning task. Evaluation on real CXR will be the next step.

## 4. Conclusion

We leveraged synthetically-generated images for learning the quantification of internal patient rotation in CXR, as originally limited by the availability of rotated and labelled CXR.

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
