# OpenReview forum: "Learning Patient Rotation Using Synthetic X-ray Images from 3D CT Volumes"
_MIDL.io/2023/Short_Paper_Track — MIDL 2023 Short paper track Poster_

### Official Review · Reviewer_fbLm · 2023-04-19
**Very clear and well illustrated paper on patient rotation estimation**

**Rating:** 8
**Confidence:** 4

**Review:**

In this article a novel training scheme is proposed that uses synthetic chest X-rays generated from 3D photon-counting CT volumes for quantifying patient rotation. This information could then be provided to the technician for possible correction. Validation on N=252 showed good correlation in angle estimation.

Pros:
* The proposed method addresses the challenge of scarcity and ethical difficulties in curating large-scale annotated clinical datasets.
* The use of synthetic data allows for a more controlled training environment.
* Results show strong correlation between true and predicted α, demonstrating the efficacy of the method.

Cons:
* Code and datasets are not provided

Ideas for related and future work:
* Evaluate the proposed method on real chest X-rays to ensure its applicability in a clinical setting.
* Investigate other potential applications of synthetic data for training neural networks in various medical image analysis tasks.

---

### Official Review · Reviewer_9vXt · 2023-04-24

**Rating:** 6
**Confidence:** 5

**Review:**

Estimating patient rotation is crucial for immobile patients. The paper proposed to use DesnetNet-121 to predict patient rotation angle from synthetic CXR images that were generated from CT volumes. Experiment results demonstrated the model was able to show a good correlation between ground truth rotation angle and predicted rotation angle. The paper is well written and organized. It is easy to read the paper. However, there are several issues: (1) descriptions of generating synthetic CXR images and  “standard post-processing and cropping” are vague, please provide related references; (2) in the experiment setup, does the data split by subjects? Any overlap between training and testing subjects? (3) how does the performance compare with other patient rotation angle estimation methods?